# Safety and Potential Role of *Lactobacillus rhamnosus* GG Administration as Monotherapy in Ulcerative Colitis Patients with Mild–Moderate Clinical Activity

**DOI:** 10.3390/microorganisms11061381

**Published:** 2023-05-24

**Authors:** Cristiano Pagnini, Maria Carla Di Paolo, Riccardo Urgesi, Lorella Pallotta, Gianfranco Fanello, Maria Giovanna Graziani, Gianfranco Delle Fave

**Affiliations:** 1Department of Gastroenterology and Digestive Endoscopy, S. Giovanni Addolorata Hospital, Via dell’Amba Aradam 9, 00184 Rome, Italy; 2Department of Gastroenterology, “Sapienza” University of Rome, 00185 Rome, Italy; 3Onlus “S. Andrea”, 00199 Rome, Italy

**Keywords:** microbiota, probiotics, inflammatory bowel diseases, ulcerative colitis

## Abstract

Probiotics are microorganisms that confer benefits to the host, and, for this reason, they have been proposed in several pathologic states. Specifically, probiotic bacteria have been investigated as a therapeutic option in ulcerative colitis (UC) patients, but clinical results are dishomogeneous. In particular, many probiotic species with different therapeutic schemes have been proposed, but no study has investigated probiotics in monotherapy in adequate trials for the induction of remission. *Lactobacillus rhamnosus* GG (LGG) is the more intensively studied probiotic and it has ideal characteristics for utilization in UC patients. The aim of the present study is to investigate the clinical efficacy and safety of LGG administration in an open trial, delivered in monotherapy at two different doses, in UC patients with mild–moderate disease. The UC patients with mild–moderate disease activity (Partial Mayo score ≥ 2) despite treatment with oral mesalamine were included. The patients stopped oral mesalamine and were followed up for one month, then were randomized to receive LGG supplement at dose of 1.2 or 2.4 × 10^10^ CFU/day for one month. At the end of the study, the clinical activity was evaluated and compared to that at the study entrance (efficacy). Adverse events were recorded (safety). The primary end-point was clinical improvement (reduction in the Partial Mayo score) and no serious adverse events, while the secondary end-points were the evaluation of different efficacies and safeties between the two doses of LGG. The patients with disease flares dropped out of the study and went back to standard therapy. The efficacy data were analyzed in an intention-to-treat (ITT) and per-protocol (PP) analysis. Out of the 76 patients included in the study, 75 started the probiotic therapy (*n* = 38 and 37 per group). In the ITT analysis, 32/76 (42%) responded to treatment, 21/76 (28%) remained stable, and 23/76 (30%) had a worsening of their clinical condition; 55 (72%) completed the treatment and were analyzed in a PP analysis: 32/55 (58%) had a clinical response, 21 (38%) remained stable, and 2 (4%) had a light worsening of their clinical condition (*p* < 0.0001). Overall, 37% of the patients had a disease remission. No severe adverse event was recorded, and only one patient stopped therapy due to obstinate constipation. No difference in the clinical efficacy and safety has been recorded between groups treated with different doses of LGG. The present prospective clinical trial demonstrates, for the first time, that LGG in monotherapy is safe and effective for the induction of remission in UC patients with mild–moderate disease activity (ClinicalTrials.gov identifier: NCT04102852).

## 1. Introduction

Inflammatory bowel diseases (IBDs) are chronic systemic conditions primarily affecting the intestine. In particular, ulcerative colitis (UC) is one of the two major forms of IBD, and it is characterized by persistent and/or recurrent inflammation of the mucosa of the colon, starting in the rectum and variably extending proximally in the colon, with the occurrence of clinical symptoms such as diarrhea, abdominal pain, bloody stools, and urgent defecation [1]. Since the etiology of the disease is unknown, no resolutive cure exists, but many different therapies, ranging from the “conventional” mesalamine and steroids, to the novel biologic drugs and small molecules, can potentially control inflammation and therefore induce and maintain the disease remission [2]. Nonetheless, a consistent proportion of patients do not respond ab initio or lose response to such therapies, with important clinical and economic implications relating to having multiple lines of treatment, access to medical care, hospitalization, and surgery [3,4]. Moreover, modern treatments are costly and not without side-effects, so novel, safe, and effective therapeutic strategies are constantly under investigation and profoundly needed.

In recent decades, a great impulse in the research of the potential interaction of the resident intestinal microbiota with the human organism both in health and in disease has pushed the concept that a “dysbiotic” microbiota may play a role in the initiation and maintenance of chronic intestinal inflammation in IBD [5]. In fact, an increased Enterobacteriaceae/Firmicutes ratio has been observed in IBD patients vs. normal controls [6], as well as an increment of “enteropathogens” with proinflammatory properties and a reduction in possible protective species (i.e., *Clostridial cluster Ⅳ* and *XIV*, *Bacteroides fragilis*, and *Faecalibacterium prausnitzii*) [7,8,9]. As a consequence of those and similar observations, the pathogenesis and the development of UC has been proposed as the result of the misbalance of the complex interactions involving microbiota, innate and acquired immune systems, and intestinal permeability. In a genetically predisposed subject, the microbiome imbalance (dysbiosis) and the loss of intestinal barrier function determine an increased antigenic pressure to the intestinal immune system, with a reduction in efficacy of the innate response and an exaggerated acquired immune system activation (with predominant Th1/Th17 lymphocytes activation and consequent production of pro-inflammatory cytokines). In a vicious cycle, the deregulated immune response increases the mucosal damage, the intestinal permeability, and the dysbiosis [10,11]. Besides their possible role in the pathogenesis, the microbial alterations have been found to be relevant in the development and in the course of the disease since specific microbial features can be associated with active inflammatory or remission states [12]. In line with these findings, the manipulation of the microbiota has been indicated as a promising field of research for potential treatments and, in particular, the administration of probiotic bacteria has been consistently investigated in IBD [13]. Probiotics are viable bacteria that, when ingested in adequate amounts, can exert beneficial effects to the host. Such bacteria may be useful as therapeutic agents in IBD exerting multiple actions. First, probiotic bacteria may contrast the dysbiosis and stimulate beneficial bacteria such as butyrate-producing species [14]. Moreover, by temporarily colonizing the intestinal mucosa and directly interacting with specific receptors of the innate immune system, probiotics may enhance the epithelial functions and survival, stimulate the mucus and anti-bacteria molecules production, reduce the intestinal permeability, and consequently decrease the antigen load to the sub-mucosal compartment, with a reduction in pro-inflammatory cytokines (i.e., TNF, IFN, and IL-17) and a stimulation of regulative mediators (i.e., IL-10, TGFb, and IL-4) [15,16,17]. Despite the fact that intense research has been conducted and a consistent amount of experimental and pre-clinical data have been produced, little or no practical clinical evidence supporting probiotic efficacy in IBD treatment has been produced. There are two main reasons for the inconsistent clinical evidence for probiotics utilization in IBD. First, because of the higher complexity of the clinical setting of IBD compared to experimental models, in which a single or a few molecular mechanisms are represented and multiple complex interactions are reduced or excluded, the multiple potential environmental and dietetic influences in real life cause IBD to be more like a syndrome comprising a wide range of different conditions hardly synthesizable in the two terms of “UC” and “CD”. Second, the studies investigating the probiotics available in the literature are characterized by an extreme dishomogeneity involving multiple aspects: patients included, numerosity, type/duration/extension of disease, study protocols, outcomes considered, and therapeutic schemes. To overcome those limitations, it is necessary to consider probiotics as fully “biotherapeutic agents” and to investigate their potential clinical application considering them just like drugs and not like dietary supplements.

Among the aforementioned dishomogeneity of the probiotic studies, the most striking evidence is the multiple bacteria species that have been investigated [18]. In fact, the term “probiotics” comprises microbial species consistently different and with peculiar species- and strain-specific properties. An ideal candidate for potential therapeutic application in UC should be a probiotic with strong safety data and well-characterized biological features, such as the adhesive capacity to the intestinal mucosa and the anti-inflammatory and immunomodulatory effect. Considering that, we intended to investigate *Lactobacillus rhamnosus GG* (LGG) that is the probiotic species more extensively investigated and characterized, has been patented since thirty years, and has a wide market distribution and a favorable safety profile, specific capacity of adhesion to the intestinal mucosa, and anti-inflammatory activity [19]. Despite specific clinical studies in IBD settings being scarce, our group has recently demonstrated, in a pre-clinical study, that LGG adheres to the colonic mucosa of UC patients in vitro and in vivo, and that it reduces the expression of pro-inflammatory cytokines, such as TNF and IL-17 [20].

In the present clinical study, the LGGinUC trial, we intended to prospectively evaluate the safety and efficacy of LGG administration in monotherapy, at two different doses, in UC patients with mild–moderate clinical activity.

## 2. Materials and Methods

### 2.1. Patients

We included consecutive UC patients followed up at the S. Giovanni Addolorata Hospital from September 2019 to January 2022. The inclusion criteria were a definite diagnosis of UC (clinical, endoscopic, and histological criteria) from at least 1 year, mild–moderate clinical activity (Clinical Mayo score 2–4) with stable symptoms in the last 6 months (chronically active disease), patient taking oral mesalamine, and informed consent obtained and signed at the screening visit (T-1). The exclusion criteria were pregnancy, serious co-morbidities (i.e., autoimmune pathologies, cancer, chronic infectious conditions, and immunocompromission), first diagnosis of UC, current immunosuppressive and/or biologic therapy for IBD, or immunosuppressive and/or biologic therapy for IBD in the last year, current oral and/or topical steroid therapy, or oral steroid therapy for disease flare in the last 6 months, current topical UC therapy (suppositories, enemas, and foams), current antibiotic/probiotic therapy, and antibiotic/probiotic therapy in the last 3 months.

### 2.2. Study Design

This study is an open randomized clinical trial that intends to evaluate the efficacy and safety of LGG (ATCC 53103) administration at two different doses, for 1 month, in UC patients with mild–moderate disease activity in therapy with oral mesalamine (Figure 1). The eligible patients were identified by performing a screening visit (T-1), with consideration of inclusion and exclusion criteria, and informed consent was signed. Then, the patients had a 4 week wash-out period, with the oral mesalamine suspended. The patients were then evaluated again prior to the randomization to a regular or double-dose group (T0), and the clinical activity was assessed. The patients were randomized to assume a regular (LGG 1.2 × 10^10^ CFU/day, 2 capsules a day) or a double (LGG 2.4 × 10^10^ CFU/day, 4 capsules a day) dose of LGG for 1 month. The LGG capsules were prepared and provided by Dicofarm (Rome, Italy) in a plain envelope with no indications or brand name on it. After 4 weeks of treatment, the patients were re-evaluated (T1), with physical examination and interview, and they were allowed to return to the therapy they were taking at the T-1 visit (mesalamine). Four weeks after treatment completion (T2), the patients were evaluated again by performing a physical examination and interview. The efficacy of the therapy was evaluated by comparing the clinical activity pre- and post-treatment (T1 vs. T0). The randomization of patients in the groups was performed by a computer-generated randomization list, in which progressive numbers, associated 1:1 to the regular or double-dose groups, were consecutively assigned to each patient. Despite the study not having a true double-blinding procedure, the clinicians and patients were unaware of their group of treatment: the patients did not know if the LGG dose they were taking was the “regular” or the “double” dose, and the investigator that assessed the parameters at the end of the treatment was different from the investigator that examined the patient at the pre-treatment visit. The clinical activity was evaluated by performing a Partial Mayo Score calculation. The clinical improvement was defined by a reduction of ≥1 point of the Partial Mayo score, and remission was defined by a Partial Mayo score ≤ 1 point. The clinical worsening was defined by an increase in the Partial Mayo score ≥ 1 point. The patients with an increase in the Partial Mayo score ≥ 2 points during the study dropped out of the study and went to specific treatment and management. In a sub-set of patients, rectosigmoidoscopy was performed before (T0) and after (T1) treatment, and the endoscopic activity was evaluated by the Endoscopic Mayo Score calculation, by an endoscopist blinded to the clinical data of the patients. The safety of the LGG treatment was assessed by weekly phone calls to the patients to investigate the unexpected occurrence of side effects, and with direct physical examination and biochemical tests (i.e., a complete blood count, serum creatinine, and transaminase) at the end of the study period (T1).

The primary end-points of the study were clinical improvement (Partial Mayo score reduction) in patients at T1 compared with T0 (efficacy), and evidence of no serious treatment-related adverse events (safety). The secondary end-points were higher clinical efficacy of double LGG dose vs. regular dose and different safety evaluations between the two groups.

The study was approved by the local ethic committee (protocol number: 0127710) and was registered to the ClincalTrials.gov web site, accessed on 25 September 2019 (Identifier: NCT04102852).

### 2.3. Statistics

Lacking specific clinical data, we calculated the sample size needed by considering the difference (of about 44%) and the standard deviation in TNF and IL-17 in vivo mucosal expression between the groups treated with a regular and high dose of LGG in a previous pre-clinical investigation [20]. Considering the α value of 0.05 and β value of 0.2 (80% power), 37 patients per group were required. This numerosity was congruous with other probiotic studies already available in the literature [21].

The efficacy data were calculated in an intention-to-treat (ITT) and per-protocol (PP) analysis by calculating the percentage of patients with improved, stable, or worse clinical disease activity at T1 compared with T0. The parametric data were compared by means of a Chi-squared test, and the non-parametric data by means of a T-test. The Clinical and Endoscopic Mayo Score before and after treatment was compared by means of a Wilcoxon Rank sum test. The uni- and multivariate analyses were performed considering several patient characteristics as binomial variables [age > 60, sex, disease location and duration > 10 years, and LGG dose (regular vs. double)], with response to LGG set as a dependent variable. A *p* value < 0.05 was considered statistically significant. MedCalc software version 12.5 was used for the statistical calculations.

## 3. Results

A total of 540 UC patients were evaluated during the study period. Of those who fulfilled inclusion criteria, 76 accepted the invitation to participate in the study and were finally included. The patients had a wash-out period of four weeks when they suspended mesalamine and were strictly followed up for potential disease worsening. One patient experienced a disease flare and went off the study and back to regular therapy, and the other 75 remained stable (Partial Mayo score unchanged) and were randomized to receive either a regular (1.2 × 10^10^ CFU/day; *n* = 38) or a double (2.4 × 10^10^ CFU/day; *n* = 37) dose of LGG. The characteristics of the included patients in the two treatment groups are presented in Table 1.

### 3.1. Efficacy

Considering the clinical outcome, only a minority of patients (30%) had a worsening of the clinical disease activity compared to that at baseline under mesalamine treatment, while 70% of the patients had either an improvement or stable condition. In particular, in the ITT analysis, 32/76 (42%) responded to treatment, 21/76 (28%) remained stable, and 23/76 (30%) had a worsening of symptoms (Figure 2B). Of the 76 patients, 21 (28%) dropped out of the study and 55 (72%) completed the treatment and were evaluated in a PP analysis (Figure 2A). Among the latter, 32/55 (58%) had a clinical response, 21 (38%) remained stable, and 2 (4%) had a light worsening of symptoms (Partial Mayo increase = 1 point) (Figure 2C). The patients who completed the study had a significant improvement in the clinical activity, and 28 patients (54%) were in clinical remission at the end of the study (Figure 2D).

In 27 patients (36%), an endoscopic examination pre- and post-LGG treatment was performed, and 7 (26%) had an improvement of their Mayo Endoscopic score, 19 (70%) remained stable, and 1 (4%) had a worsening of the score (Figure 3A). In the paired data analysis, the endoscopic score significantly improved after treatment with LGG (Figure 3B).

Comparing the two different doses of LGG, no significant difference emerged, and the patients in both groups had a significant clinical response compared to T0 (Figure 4A–C). No clinical characteristic nor LGG dose taken was associated with a response to treatment in the uni- and multivariate analyses.

### 3.2. Safety

No serious adverse event was recorded during the study. Fifteen patients (20%) reported adverse events, which were mild and spontaneously resolved in all but one patient (1.3%) who had to stop the therapy due to obstinate constipation. The adverse events are reported in detail in Table 2.

One patient dropped out due to a flare in the wash-out phase of the study. Among the 75 patients who started the treatment, the therapy was stopped mainly for disease flare [20/75 (27%) patients], and for the aforementioned adverse event (constipation in one patient). Furthermore, 8/20 (40%) of the disease flare occurred during the first week of treatment, 9 (45%) after two weeks, and 3 (15%) after three weeks of treatment. At the end of the study period, all the patients resumed mesalamine therapy.

## 4. Discussion

The present study, in an attempt to overcome the limitations of the previously published probiotic studies in IBD settings, investigated the potential clinical application of a well-characterized probiotic species, namely LGG, as a treatment in mild–moderate UC patients. LGG administration has been proven to be safe both in the regular and in the double-dose groups. Considering the clinical effect, the chronically active UC patients that assumed LGG as the sole therapy had a satisfactory response, and only a minority of patients had a disease flare during the study.

The study presents some points of strength. First, it is monocentric, so the heterogeneity in the selection of patients and in the methods of follow-up is reduced. Second, it included a homogenous set of patients with defined clinical characteristics (chronically active disease with mild–moderate activity despite treatment with oral mesalamine) and with a congruous number, which allowed for a fair evaluation of the proposed end-points. Moreover, the primary and secondary end-points were clearly stated. Besides the novel outcomes for treatment evaluation being proposed (i.e., biochemical parameters, endoscopic, and histological healing) [22], the main reference in trials and clinical practice remains the clinical response, which we quantified using the easy and accurate Partial Mayo score. Although not present in the study end-points, we also included the endoscopic activity in the evaluation, and, indeed, a significant trend for the amelioration of the endoscopic score was observed after LGG treatment. Unfortunately, due to the endoscopic restriction because of the COVID-19 pandemic, we were only able to perform pre- and post-endoscopic evaluation in a small subset of patients (36%), so the results are not conclusive.

Indeed, in the present study, we chose to investigate a single-strain formulation, with a probiotic bacteria that has been extensively studied and investigated in different inflammatory conditions, and with a strong safety background and ideal features for therapeutic application in UC patients. In fact, LGG has a specific adhesive capacity to the intestinal epithelium for the presence of pili and for the secretion of mucus-binding proteins [23,24]. Moreover, it induces epithelial protection and normalization of intestinal permeability, by means of NF-κB pathway modulation, production of soluble proteins with protective anti-bacterial effects (i.e., p40, p75, and mucins), and biofilm formation [25,26,27,28]. Finally, LGG exerts immunomodulatory activity with stimulation of the innate response and reduction in the pro-inflammatory acquired response [29,30], and promotes dysbiosis reduction and stimulation of butyrate-producing bacteria [31,32]. In clinical settings, LGG administration has proven beneficial in infective and antibiotic-related diarrhea [33,34], respiratory tract infections [35], and allergic diseases [36]. Besides having such a consistent amount of data, LGG has not yet been adequately investigated in IBD patients. Zocco et al. demonstrated that LGG administration was equally effective to mesalamine in remission maintenance in UC patients, with a similar relapse rate at 6 and 12 months [37]. Tong et al. have recently demonstrated that LGG extracellular vesicles administration is effective in reducing inflammation in DSS-induced colitis by means of inhibition of the activation of the TLR4-NF-κB-NLRP3 axis and the consequent reduction in proinflammatory cytokines and dysbiosis correction [38]. In a pre-clinical study, our group has already demonstrated that LGG effectively adheres to the colonic mucosa and reduces the expression of pro-inflammatory cytokines (i.e., TNF and IL-17), in vitro and in vivo, and that a higher LGG dose showed a more marked effect [20]. In order to bring such results into a clinical setting, in the present study, we investigated the clinical effect of LGG administration at two different doses in UC patients with mild moderate chronic activity.

As already mentioned in the introduction, the clinical data available for probiotic utilization in IBD are profoundly dishomogenous and do not allow for definitive conclusions nor clinical indications. In particular, very recently, a Cochrane review analyzed the effectiveness of probiotics for remission induction in UC patients. The authors identified 14 eligible studies, and, among those, 9 investigated the efficacy of probiotics vs. placebo in RCT trials (including a total of 594 patients), with a slight superiority of probiotics in inducing remission (RR 1.73 and NNT 5), but with low-certainty evidence and consistent differences among the studies [21]. Most importantly, all the available studies investigated the probiotic administration as an add-on therapy, allowing concomitant treatments (i.e., oral mesalamine, prednisolone, and immunosuppressant). In order to fully evaluate probiotic efficacy, for the first time, in the present study, LGG was investigated for the induction of remission in monotherapy with no other concomitant UC treatment allowed during the study. Moreover, as a further factor to avoid possible confounding factors, we chose to adopt a consistent wash-out phase (i.e., one month), with the suspension of oral mesalamine, before the starting of LGG treatment. Considering that, we included patients with mild–moderate activity of disease that were clinically stable (stable symptoms and no steroid use in the last 6 months, no topical treatment, and no recent disease worsening). Nonetheless, since most of the disease flares were observed in the present study in the first two weeks of treatment, we cannot exclude that they were related to the long therapy-free period. It would be interesting to investigate the clinical efficacy of a longer treatment with LGG in patients with more severe disease and with a shorter duration of mesalamine interruption.

One open issue in the probiotic studies is the duration of the treatment, and, in fact, the studies’ durations in the previously quoted Cochrane meta-analysis ranged from 2 to 52 weeks [21]. In a pre-clinical study, we found that LGG adheres to the colonic mucosa and inhibits the pro-inflammatory cytokines as early as after one week of administration [20]. In the present study, we proposed a probiotic treatment for one month. Considering the chronic characteristic of the disease, and the fact that we included mainly UC patients with mild disease, a longer treatment period would probably lead to even more remarkable results. Considering the difference in adhesion and the anti-inflammatory effect found in our previous paper between two different doses of LGG, we intended to evaluate the clinical effect of the two different doses in UC patients. Indeed, in the present study, no difference has been found in the safety and efficacy in patients treated with the two doses of LGG, which probably further confirms the difference in experimental vs. clinical real-life studies. Nonetheless, bearing in mind the previous consideration about the setting of the included patients (chronically active disease mainly with mild activity) and the duration of treatment, the potential dose-related effect of LGG needs further molecular and clinical investigation.

The present study has several limitations. First of all, it is an uncontrolled study. Randomized clinical trials (RCT) remain the mainstay for the evaluation of therapeutic options, and, therefore, the lack of a control group in the present study consistently limits the evaluation of the treatment efficacy. Notwithstanding that, the investigation of a probiotic formulation in adequate numerous studies with clear end-points may still represent a contribution to this field of research. Moreover, we had a screening phase (with no UC medication allowed) that lasted as the treatment phase (i.e., one month). Therefore, we had a sort of internal “control group” for the patients before starting LGG therapy. In that period, all the patients but one remained clinically stable, while, after the treatment period, most of the patients had a clinical improvement (42%) and more than two-thirds of the patients finished the study with stable or better clinical disease activity compared to that at the study entrance, with 37% reaching disease remission (Partial Mayo score ≤ 1). Considering that previous studies have never investigated probiotics as monotherapy for the induction of remission of UC patients, we chose to adopt two treatment groups (with different doses) in this study, but, in line with our results, the next step should be the design of an RCT with LGG in monotherapy with a placebo group. Another limitation of the study is that, lacking specific clinical data, and since the present studies did not include a placebo control group, we used the difference in the mucosal expression of pro-inflammatory cytokines between two groups treated with different doses of LGG for the sample size calculation. Indeed, the translation of such a molecular effect to the actual clinical symptom’s improvement is not straightforward, and more precise clinical data are needed to correctly design adequately powered studies to evaluate the potential different efficacies of probiotic doses. Anyway, the sample size considered was comparable to that of most of the probiotic studies already available in the literature [21].

A molecular analysis of the potential mucosal effect of LGG and of the quantitative and qualitative microbiome variations was out of the aim of the present paper. Nonetheless, the preliminary analysis of the available bioptic samples in patients after LGG administration, by means of DNA extraction and quantification using Real Time (RT)-PCR, showed that LGG DNA is constantly detectable at the mucosal level. The evaluation of the modulation of pro- and anti-inflammatory mediators at the mucosal level after LGG administration is ongoing. Zmora et al. have recently demonstrated that probiotic colonization may depend on the individual microbiota characteristics, which may be “permissive” or “resistant” to exogenous colonization [39]. It would be interesting to evaluate the basal microbiome characteristics in the patients of the present studies to evaluate the specific features associated with the clinical response to LGG and, therefore, the molecular predictors of probiotic therapy efficacy, as well as the potential modification induced in the microbiota by LGG administration.

In conclusion, we demonstrated, for the first time, that LGG administration in monotherapy may be a safe and feasible option for the induction of remission in UC patients with chronically active mild–moderate disease activity. Probiotic therapy could be a possible option in UC patients, and many aspects still deserve investigation. The identification of specific probiotic bacteria, the clinical investigation of precise outcomes in selected groups of patients, and the research of the molecular effect and microbiota modulation of such identified species, minimizing the potential confounding factors such as concomitant therapies, would lead to a real advance in the research and clinical application of probiotics as biotherapeutic agents in IBD patients.

## Figures and Tables

**Figure 1 microorganisms-11-01381-f001:**
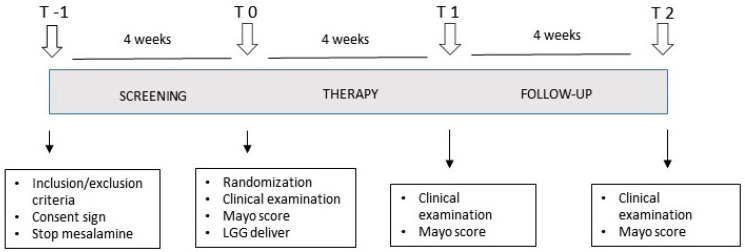
Schematic representation of the study outline. The study comprised four visits for the patients: a pre-screening evaluation (T-1), the starting visit after 4 weeks of therapy wash-out (T0), the final visit at the end of the treatment period (T1), and a follow up visit one month after the end of the study (T2).

**Figure 2 microorganisms-11-01381-f002:**
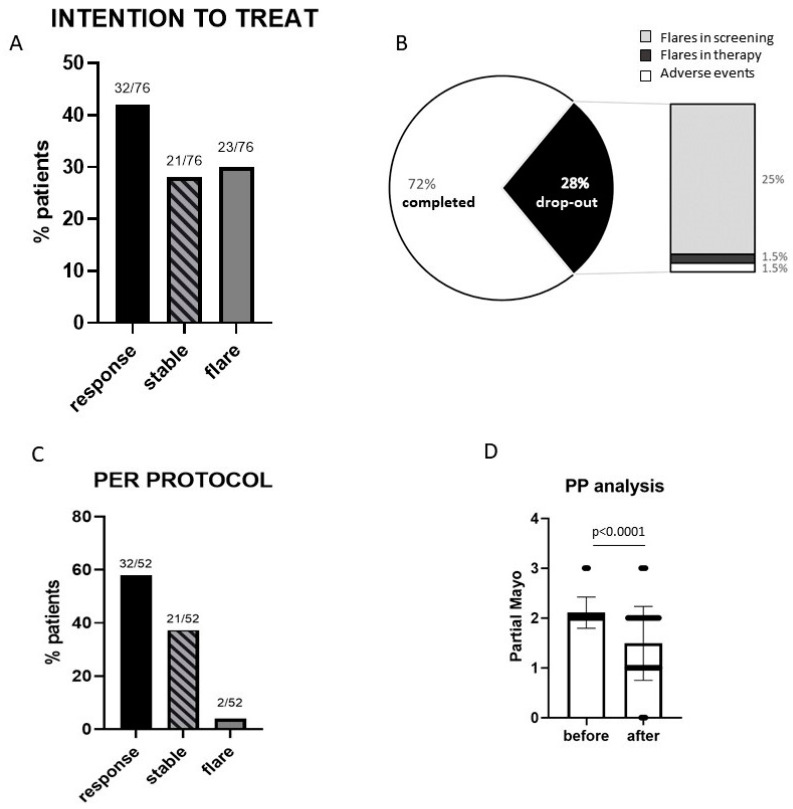
Clinical effect of LGG administration in UC patients. (**A**) Intention-to-treat analysis: comparison of clinical activity at the end of the treatment period (T1 vs. T0 visit) of the total of the patients who started the treatment protocol (*n* = 76). (**B**) Proportion of patients who finished the study (further analyzed in the per-protocol analysis) and who dropped out (the stacked column indicates the reason and the relative proportion for the study exit). (**C**) Per-protocol analysis: comparison of clinical activity at the end of the treatment period (T1 vs. T0 visit) of the subset of patients who completed the treatment protocol (*n* = 52). (**D**) Representation of the comparison of the clinical activity before and after probiotic treatment (T1 vs. T0 visit) for each patient who completed the study. Statistical comparison was performed using a Wilcoxon test for paired samples.

**Figure 3 microorganisms-11-01381-f003:**
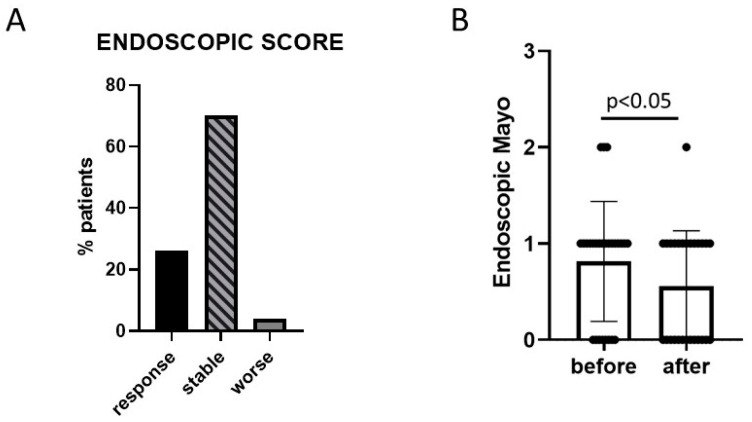
Effect on endoscopic activity of LGG administration. (**A**) Comparison of endoscopic activity at the end of the treatment period (T1 vs. T0) of the subset of patients with pre- and post-treatment endoscopic examination (*n* = 27, 36% of patients). (**B**) Representation of the comparison of the endoscopic activity before and after probiotic treatment (T1 vs. T0). Statistical comparison was performed using a Wilcoxon test for paired samples.

**Figure 4 microorganisms-11-01381-f004:**
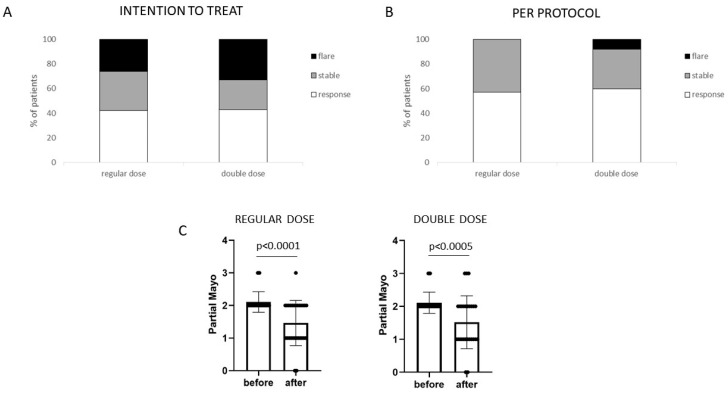
Clinical effect of LGG administration in two different doses (regular = 1.2 × 10^10^ CFU/day; double dose = 2.4 × 10^10^ CFU/day). No significant difference was recorded between the two groups. (**A**) Intention-to-treat analysis: comparison of clinical activity at the end of the treatment period (T1 vs. T0 visit) in the patients who started the treatment protocol in regular and double-dose group (*n* = 38 and 37, respectively). (**B**) Per-protocol analysis: comparison of clinical activity at the end of the treatment period (T1 vs. T0 visit) in the patients who completed the treatment protocol in regular and double-dose group (*n* = 28 and 27, respectively). (**C**) Representation of the comparison of the clinical activity, in regular and double-dose group, before and after probiotic treatment (T1 vs. T0 visit), in patients who completed the treatment protocol. Statistical comparison was performed using a Wilcoxon test for paired samples.

**Table 1 microorganisms-11-01381-t001:** Characteristics of the UC patients included in the study.

Characteristics	Regular Dose (*n* = 38)	Double Dose (*n* = 37)	Total (*n* = 76)
Age (years)	54 ± 14	60 ± 14 *	57 ± 15
Gender (M/F)	15/23	21/16	37/39
Disease duration (years)	9.5 ± 8	9.8 ± 6.2	9.6 ± 7.1
Extension:			
Proctitis	6 (16%)	6 (17%)	12 (16%)
Proctosigmoiditis	21 (55%)	19 (51%)	41 (54%)
Pancolitis	11 (29%)	12 (32%)	23 (30%)
Partial Mayo			
2	31 (82%)	30 (81%)	62 (82%)
3	4 (10%)	6 (16%)	10 (13%)
4	3 (6%)	1 (3%)	4 (5%)

* = *p* < 0.05 vs. regular dose group.

**Table 2 microorganisms-11-01381-t002:** Adverse events (AEs) observed in the LGGinUC trial.

Adverse Event (AE)	Frequency
Disease flare	24/76 (32%)
Bloating	9/76 (12%)
Constipation	4/76 (5%)
Abdominal pain	2/76 (3%)
Headache	1/76 (1%)

## Data Availability

Data available upon request to the corresponding author.

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
