# Peer review of "Safety and Potential Role of Lactobacillus rhamnosus GG Administration as Monotherapy in Ulcerative Colitis Patients with Mild–Moderate Clinical Activity"

_microorganisms, 2023, doi:10.3390/microorganisms11061381_

Round 1
Reviewer 1 Report (Previous Reviewer 2)
Authors corrected the manuscript accroding my comments.
Author Response
We thank the reviewer for the positive comment on our work
Reviewer 2 Report (Previous Reviewer 1)
Authors have addressed most of my comments.
Some grammar editing is still required.
Author Response
We thank the reviewer for the positive comment. We doublechecked the paper for grammar editing with English expert. We highlited the changes with the track changing function.
This manuscript is a resubmission of an earlier submission. The following is a list of the peer review reports and author responses from that submission.
Round 1
Reviewer 1 Report
Major Comments
The introduction would benefit from discussion of the microbiota of UC and why specifically LGG is the ideal probiotic choice. More references are required to support statements in the introduction. Discussion claims this is the first study to use monotherapy of a probiotic in UC and this is entirely false, evidenced by the reference included by the authors. The manuscript would benefit from professional grammar editing, I have highlighted a few changes to be made below but did not have the time to correct the full manuscript (especially when line numbers are not provided).
Specific comments
Abstract:
· remove “among those,” from 2nd sentence
· , but no study “has” investigated
· “The” aim of the present study
· one patient “had” a disease flare
· I would recommend including p-values in the abstract when reporting all results. Example: “Only a minority of patients had a worsening of disease activity (30%; “p= xxxx”). In addition, the following sentence to this the authors repeat the values for worsening of clinical condition, if this is a repeat please remove or rephrase to differentiate from the previous sentence.
Introduction:
· edit sentence to match “can potentially control inflammation” and remove “obtain” etc
· “Nonetheless, a consistent”
· Microbial genus and species names should be in italics
· “administration of probiotic bacteria has been consistently investigated in IBD.” Provide references for this statement.
· “butyrate” not butirrate
· “no practical” not “none practical”
· “IBD appears more like a syndrome comprising a wide range” not “the same IBD appears more and more like a syndrome comprising a wide range”
· “Among the aforementioned dishomogeneity of the probiotic studies, the most striking is the multiple bacteria species investigated” add references
· “scarce” not “scares”
Methods
· Provide details on where the LGG capsules were sourced or how these were prepared.
· Provide details on which computational randomization tool was used specifically
· How can authors claim that patients or clinicians weren’t aware that they were receiving the regular or double dose when the number of tablets were different between the 2 groups, 4 tablets would easily be identified as the double dose if authors were using a commercial product and the recommended dosage is 1-2 for most probiotic products. For true blinding the number of tablets should have been consistent between the groups (with placebos included to control for the difference in tablets).
· Use past tense consistently here and through the manuscript.
Results
· “About 540” can authors please be specific and report the actual number of patients evaluated or remove the word “about” if this is the exact number.
· Same comment from abstract, p-values also need to be included when reporting % of outcomes in results.
· LGG dose “consumed” not “assumed”
· How many patients resumed mesalamine following completion of LGG therapy?
· Figure 2 and 3 include statistical comparisons between groups for Figure 2A, 2C, 3A
· Actual descriptive figure legends are required to describe the results and analyses.
· Table 1. What does the asterisk mean? Include this information in the table legend. Stats on each of the reported characteristics should also be included in this table.
Discussion
· Reference Zocco (2006) seems to have already conducted this study (with a slightly lower dose over a much longer period) and concluded that LGG is effective and maintaining remission of UC, what novelty does this study bring to the field? I understand that they wanted to test a higher dose to reduce the expression of pro-inflammatory cytokines, but authors do not report any of these host factors. Nor have authors included a control group that was taken off mesalamine and provided no probiotic treatment for the duration of the trial, I struggle to see how conclusions can be drawn around the rate of remission/reduction of symptoms induced by LGG without these extra measures over such a short period of time (and the only patient to experience a flare in the lead up removed from the trial). Really the only novel conclusion is that a LGG dose of 2.4 × 10^10 CFU was well tolerated by UC patients.
· Authors claim this is the “first study” to test LGG as a solo treatment but their own reference to Zocco did the same (LGG alone, mesalamine alone, and LGG + mesalamine).
· “Pretty long” what evidence do authors have for selecting this washout period, please use more precise scientific language.
· “since the most of the disease flares were observed in the first two weeks of treatment” is this an observation from the study by authors (they only reported 1 flare) or is this from literature? If so provide references.
· Authors elude to qPCR results of host factors but do not report data, I think the manuscript would benefit from these findings to provide a more comprehensive addition to literature.
